# *BRAF* Mutation Status in Circulating Tumor DNA from Patients with Metastatic Colorectal Cancer: Extended Mutation Analysis from the AGEO RASANC Study

**DOI:** 10.3390/cancers11070998

**Published:** 2019-07-17

**Authors:** Leo Mas, Jean-Baptiste Bachet, Valerie Taly, Olivier Bouché, Julien Taieb, Romain Cohen, Aurelia Meurisse, Corinne Normand, Jean-Marc Gornet, Pascal Artru, Samy Louafi, Anne Thirot-Bidault, Isabelle Baumgaertner, Romain Coriat, David Tougeron, Thierry Lecomte, Florence Mary, Thomas Aparicio, Lysiane Marthey, Helene Blons, Dewi Vernerey, Pierre Laurent-Puig

**Affiliations:** 1Department of Hepato-Gastroenterology, Groupe Hospitalier Pitié Salpêtrière, 75013 Paris, France; 2AGEO (Association des Gastroentérologues Oncologues), 75013 Paris, France; 3Sorbonne Université, UPMC Université, 75006 Paris, France; 4Centre de Recherche des Cordeliers, INSERM, Sorbonne Université, USPC, Université Paris Descartes, Université Paris Diderot, Team MEPPOT, CNRS SNC 5014, Equipe labellisée Ligue Nationale Contre le Cancer, 75006 Paris, France; 5Department of Hepato-Gastroenterology, Hôpital Robert Debré, 51100 Reims, France; 6Department of Oncology Digestive, Hôpital Européen Georges Pompidou, 75015 Paris, France; 7Department of Oncology, Hôpital Saint-Antoine, 75012 Paris, France; 8Department of Methodology and Quality of Life in Oncology, INSERM UMR 1098, Hôpital Universitaire de Besancon, 25000 Besancon, France; 9Department of Gastroenterology, Hôpital Saint-Louis, 75010 Paris, France; 10Department of Gastroenterology, Hôpital Privé Jean Mermoz, 69008 Lyon, France; 11Department of Gastroenterology, Centre Hospitalier Sud Francilien, 91100 Corbeil-Essonnes, France; 12Department of Gastroenterology, Groupe Hospitalier Nord Essonne, 91160 Longjumeau, France; 13Department of Gastroenterology, Hôpital Kremlin Bicêtre, 94043 Le Kremlin-Bicêtre, France; 14Department of Oncology, Hôpital Henri Mondor, 94000 Créteil, France; 15Department of Gastroenterology, Hôpital Cochin, 75014 Paris, France; 16Depatment of Gastroenterology, Centre Hospitalo-Universitaire de Poitiers, 86000 Poitiers, France; 17Department of Gastroenterology, Centre Hospitalo-Universitaire de Tours, 37000 Tours, France; 18Department of Gastroenterology, Hôpital Avicenne, 93000 Bobigny, France; 19Depatment of Gastroenterology, Hôpital Antoine Béclère, 92023 Clamart, France; 20Department of Biochemistry, Hôpital Européen Georges Pompidou, 75015 Paris, France

**Keywords:** colorectal cancer, circulating tumor DNA, accuracy, liver metastases, NGS, methylated biomarker

## Abstract

In patients with metastatic colorectal cancer (mCRC), *RAS* and *BRAF* mutations are currently determined by tumor sample analysis. Here, we report *BRAF* mutation status analysis in paired tumor tissue and plasma samples of mCRC patients included in the AGEO RASANC prospective cohort study. Four hundred and twenty-five patients were enrolled. Plasma samples were analyzed by next-generation sequencing (NGS). When no mutation was identified, we used two methylated specific biomarkers (digital droplet PCR) to determine the presence or absence of circulating tumor DNA (ctDNA). Patients with conclusive ctDNA results were defined as those with at least one mutation or one methylated biomarker. The kappa coefficient and accuracy were 0.79 (95% CI: 0.67–0.91) and 97.3% (95% CI: 95.2–98.6%) between the *BRAF* status in plasma and tissue for patients with available paired samples (*n* = 405), and 0.89 (95% CI: 0.80–0.99) and 98.5% (95% CI: 96.4–99.5%) for those with conclusive ctDNA (*n* = 323). The absence of liver metastasis was the main factor associated to inconclusive ctDNA results. In patients with liver metastasis, the kappa coefficient was 0.91 (95% CI, 0.81–1.00) and accuracy was 98.6% (95% CI, 96.5–99.6%). We demonstrate satisfying concordance between tissue and plasma *BRAF* mutation detection, especially in patients with liver metastasis, arguing for plasma ctDNA testing for routine *BRAF* mutation analysis in these patients.

## 1. Introduction

Colorectal cancer (CRC) is the second most common cancer worldwide and the second leading cause of cancer death in Europe with about 500,000 new cases and 242,000 deaths in 2018 [1]. Molecular diagnostics has become a key element of the management of metastatic CRC (mCRC). Besides the well-established *RAS* testing used to determine the efficacy of anti-EGFR therapies, *BRAF* mutation status represents a biomarker of growing interest.

*BRAF* mutations occur in approximately 10% of mCRCs with *BRAF* V600E accounting for 80% to 90% of cases [2,3,4,5]. *BRAF*-mutated mCRC show distinct clinical features and poorer survival than *BRAF* wild-type cancers [6,7]. Despite the lack of phase III randomized trials of *BRAF*-mutated mCRC, a distinct therapeutic strategy is recommended for these patients including a first-line treatment with the triplet chemotherapy combination of fluoropyrimidines (folinic acid, 5-fluorouracil), oxaliplatin and irinotecan (FOLFOXIRI) and bevacizumab (an anti-vascular endothelial growth factor (VEGF) agent) [8,9]. Targeting the *BRAF* mutation represents a promising new therapeutic approach. There have been disappointing phase I/II trials results showing that a BRAF inhibitor monotherapy (vemurafenib, encorafenib) did not demonstrate any valuable clinical efficacy [10,11]. However, combination strategies with anti-EGFR and MEK inhibitors to overcome primary resistance to BRAF inhibition in mCRC have been successful [12,13,14]. The results from the phase III BEACON trial showed that the triplet regimen of encorafenib, binimetinib and cetuximab led to the objective response rate of 48% and was associated with a median progression-free survival of 8.0 months and a median overall survival of 15.3 months, in 29 pre-treated patients with *BRAF* V600E-mutant mCRC [15]. Currently, *RAS* and *BRAF* mutational testing is carried out from tumor tissue. However, it is a complex process involving many healthcare specialists, which leads to delayed test results, up to one month in France for instance [16]. This aspect is specifically of concern in a situation when patients need urgent treatment and/or need to be included in first-line clinical trial.

Liquid biopsy is a minimally invasive approach allowing us to detect and analyze molecular biomarkers from peripheral blood such as circulating tumor DNA (ctDNA) in cancer patients. In mCRC, ctDNA has been evaluated as a prognostic marker and a potential tool to guide adjuvant therapy, monitor treatment responses and identify the occurrence of resistance mutations in the metastatic setting [17,18,19,20,21]. In addition, ctDNA testing unlike tissue analysis is a quick and non-invasive procedure. Douillard et al. have demonstrated that analysis of ctDNA is able to detect accurately *EGFR* mutation status in non-small-cell lung cancer [22]. However, robust evidence of concordance between tissue and plasma analysis is still needed before this approach can be used for mCRC patients in clinical practice.

In the AGEO RASANC large prospective multicenter study, plasma *RAS* mutation detection has been directly compared with tumor tissue in 412 mCRC patients [23]. These data showed a very high overall concordance in 329 patients with detectable ctDNA with the kappa coefficient of 0.89 and accuracy of 94.8%. Here we report updated results from the AGEO RASANC study extended to *BRAF* mutation analysis in mCRC patients. We explored the concordance between the mutational status of the *RAS* and *BRAF* in tumor tissue and plasma.

## 2. Results

### 2.1. Study Population

Overall, 425 patients were enrolled in the AGEO RASANC prospective cohort study in 14 French centers between July 2015 and December 2016. Patients who did not meet the inclusion criteria, had no available tumor tissue or plasma samples (*n* = 13), and did not have *BRAF* assessment (*n* = 5) were excluded, leaving 407 patients for analysis.

#### BRAF and RAS Mutation Status

Of the 407 patients, 7.4% (30/406) and 6.6% (27/406) had *BRAF* mutations in tumor tissue and plasma, respectively.

In total, 33 patients with *RAS-*mutated tumors were not assessed for *BRAF* mutation status; these were considered as *BRAF* wild-type patients.

Only one patient had both *RAS* and *BRAF* (other than the V600E) mutations. This patient was classified as being positive for a *BRAF* mutation for all analyses.

### 2.2. ctDNA Detection

Two patients did not have paired tumor tissue and plasma to determine the level of concordance of *BRAF* mutation and were excluded from the analyses (one tissue and one plasma *BRAF* mutational status unavailable).

Of the 405 (99.5%) patients included in the concordance analysis, 323 (79.7%) had conclusive ctDNA results (at least one mutated gene and/or one methylated biomarker identified) and 82 (20.2%) had inconclusive ctDNA results (the absence of the mutated gene or methylated biomarker).

#### 2.2.1. Concordance of the BRAF Mutation Between Tissue and Plasma

In the overall population tested for *BRAF* mutations in both tumor tissue and plasma (*n* = 405), the kappa coefficient was 0.79 (95% CI: 0.67–0.91), the accuracy was 97.3% (95% CI: 95.2–98.6%), the sensitivity was 76.7% (95% CI: 57.7–90.1%), and the specificity was 98.9% (95% CI: 97.3–99.7%; Table 1).

In the 323 (79.7%) patients with conclusive ctDNA results, the kappa coefficient was 0.89 (95% CI: 0.80–0.99), the accuracy was 98.5% (95% CI: 96.4–99.5%), the sensitivity was 95.8% (95% CI: 78.9–100.0%), and the specificity was 98.7% (95% CI: 96.6–99.6%; Table 1).

#### 2.2.2. Discordant Cases

In 5.7% of patients (23/405), mutations were detected in both tumor tissue and plasma, while 1.7% (7/405) had mutations only in tissue and 1.0% (4/405) had mutations only in plasma.

Of the seven patients with the *BRAF* mutation in tumor tissue, but not in paired plasma, one had conclusive ctDNA results as assessed by the presence of a *TP53* mutation (the mutation allelic frequency was 3%) and six had inconclusive ctDNA result with neither gene mutation nor biomarker methylation identified.

Of the four patients with the *BRAF* mutation in plasma, but not in the paired tumor tissue, two were positive for a *BRAF* non-V600E mutation (one p.Asp594His and one p.Lys601Glu mutation). Tumor tissues screening by PCR PNA SNaPshot or Sequenom mass spectrometry-based assay of these later patients did not reveal *BRAF* non-V600E mutations. The second round of tumor tissue analysis by next-generation sequencing (NGS) was possible in only two out of four patients with the *BRAF* mutation in plasma. One tumor harbored the *BRAF* mutation (p.Asp594His mutation). Poor DNA quality of the tumor sample did not allow us to assess the *BRAF* mutation status in the second patient.

#### 2.2.3. BRAF Mutated Patients with Conclusive and Inconclusive ctDNA Results

Of the 34 patients with *BRAF* mutated tumors, 28 (82.3%) had conclusive ctDNA results and six (17.6%) had an inconclusive ctDNA result. Characteristics of these patients are summarized in Table 2.

Primary tumor resection, the absence of liver metastases and peritoneal carcinomatosis were significantly more frequent among patients with inconclusive ctDNA results. These patients had significantly lower alkaline phosphatase and carcinoembryonic antigen (CEA) levels (Table 2).

#### 2.2.4. Concordance of BRAF Mutation Status According to the Presence of Liver Metastases and/or Primitive Tumor Resection

In the patients with liver metastases tested for *BRAF* mutations in both the tumor tissue and plasma (*n* = 405), the kappa coefficient was 0.91 (95% CI: 0.81–1.00) and the accuracy was 98.6% (95% CI: 96.5–99.6%). In patients with liver metastases and unresected primary tumor, the kappa coefficient was 0.86 (95% CI: 0.73–0.99) and the accuracy was 97.5% (95% CI: 93.8–99.3%).

Similar results were obtained in patients with conclusive ctDNA results (*n* = 323; Table 3).

### 2.3. Comparison of Baseline Patient Characteristics According to RAS/BRAF Status

Of the 405 patients, 34 (8.4%) and 244 (59.9%) were found to have *BRAF* and *RAS* mutation, respectively. There were 129 (31.6%) patients in whom neither a *BRAF* nor *RAS* mutation was detected (wild-type patients).

#### 2.3.1. Clinical and Pathological Data at Baseline

The clinical and pathological characteristics of the 407 patients are presented in Table 4. *BRAF* mutated tumors were significantly more right-sided, undifferentiated and had more infra and supradiaphragmatic adenopathy. *BRAF* mutation occurred more frequently in younger and females patients, although this association was not statistically significant. No significant differences were also observed in ECOG PS and T/N stages between *BRAF* mutated and wild-type tumors (Table 4).

*BRAF* mutated or *RAS* mutated patients showed a distinct metastatic pattern with more synchronous metastases and fewer resectable metastases than those with wild-type tumors.

#### 2.3.2. Biological Features

CEA levels were significantly higher in *BRAF* or *RAS* mutated patients than in those with wild-type tumors. *BRAF* mutated patients had significantly higher CA 19.9, lactate dehydrogenase and alkaline phosphatase levels than those with *RAS* mutated or wild-type tumors. Moreover, a trend toward a lower lymphocyte count was observed in those patients (Table 4).

#### 2.3.3. Microsatellite Instability (MSI) Status

For 133 (32.6%) patients, microsatellite instability (MSI) status was available; including 29 of the 30 patients with tumor tissue *BRAF* mutation and 26 of the 27 patients with *BRAF* plasma mutation. Only 10 (7.5%) patients harbored MSI; five of them had *BRAF* mutation in both tumor tissue and plasma, one had a mutation only in tissue, and one had mutation only in plasma.

*BRAF* mutation both in the tumor tissue and plasma was strongly associated with the presence of the MSI status (Table 5). MSI patients were more frequently females (80.0% vs. 45.5%; *p* = 0.0486) and had significantly more right-sided tumors (70.0% vs. 29.0%; *p* = 0.0168) than microsatellite stability (MSS) patients.

## 3. Discussion 

The primary report of the AGEO RASANC study demonstrated an excellent concordance between *RAS* mutation status in tumor tissue and plasma of untreated mCRC patients. Here, we extend these findings to *BRAF* mutation analysis and demonstrated a high level of concordance between tissue and plasma with the accuracy reaching 97.3% in the overall population and 98.5% in patients with conclusive ctDNA results (kappa index of 0.89).

The sensitivity and specificity of ctDNA *BRAF* mutation detection for the patients with conclusive ctDNA results were 95.8% and 98.7%, respectively. These rates are similar to those reported on *RAS* mutation detection in the present cohort [23] and to several previous studies [24,25]. Limited data are available on the *BRAF* mutation detection in this setting. Van Cutsem et al. evaluated the concordance of *RAS/BRAF* mutation status in tumor issue and plasma of patients with mCRC. In total, 70 patients had paired tumor and plasma available for *BRAF* V600E mutation status analysis. The prevalence of *BRAF* V600E mutation in plasma was 11.4% versus 12.9% in tumor tissue, resulting in the concordance of 95.7% (kappa coefficient of 0.80) [26]. In a prospective study by Thierry et al. [27] mutations were determined by the IntPlex® PCR method from 97 mCRC patient samples. ctDNA analysis showed low sensitivity (57%) and specificity (89%) for the *BRAF* V600E mutation. In the study by Kato et al. [28] NGS of ctDNA (the foundation one test was performed on tissue samples) was performed in 94 patients with CRC. They reported the sensitivity of 80% and the concordance between tissue and plasma NGS of 85.5% for *BRAF* (kappa coefficient of 0.512). The concordance for patients with *BRAF* V600E mutation only (*n* = 8) was 100%.

In the RASANC study, 20% of patients had inconclusive results due to the absence of both mutations by NGS and the methylation marker by digital droplet PCR, hindering the distinction between the absence of ctDNA or the mutation of interest. This proportion is in line with that reported in previous studies [17].

In the current study, seven patients harbored a *BRAF* mutation only in the tissue. In six of them this discordant result might be explained by inconclusive results for ctDNA analysis. One patient was positive for ctDNA detection by NGS with the allele frequency of 3%.

Both the presence of liver metastases and unresected primitive tumor were significantly associated with the ctDNA detection, but the presence of liver metastases appeared to be the primary factor. This observation is in line with our previous report on *RAS* mutations for this cohort [23]. The concordance rate for *BRAF* mutation in patients with liver metastases was 98.6%, without any discordant case with tissue mutation undetected in plasma. The capacity to assess the presence or absence of ctDNA appears to be a critical issue for high accuracy molecular testing in plasma; methods that is only possible to be used in patients with detectable ctDNA. The mechanism of metastatic diffusion (hematogenous versus other) seems to be a major factor associated to the presence of ctDNA, explaining that accuracy is very high in patients with liver metastases, but not in those with peritoneal carcinomatosis. According to these results, assessment of *RAS* and *BRAF* status through ctDNA in patients with mCRC and liver metastases appears to be a highly sensitive method. Such an approach could be used to reduce the delay for obtaining molecular results, but also to monitor the treatments efficacy [17,18,19]. In such a way, reductions in the *BRAF* V600E ctDNA allele fraction have been shown to predict a radiographic response after treatment with MAPK pathway targeting agents in phase Ib and I/II studies [13,29].

Four patients in our study had *BRAF* mutation only in plasma. In three of these discordant cases negative tissue results could be the result of sampling error in a tumor with intra-tumoral heterogeneity (among one or different tumor sites) of mutant *BRAF* expression [19]. As ctDNA is supposed to be representative of the global tumor burden, the plasma tumoral status detection in those discordant cases might be even more efficient than tissue analysis focusing on a small sample of tumor cells.

Two patients in the current study had *BRAF* non-V600E mutation only in plasma; this is probably because these mutations may not be detectable by routine tissue analysis method, highlighting another potential gain from plasma-based NGS in clinical practice.

Indeed, *BRAF* non-V600E mCRC have been reported as a distinct subgroup of patients with specific clinical and pathological features (younger patients, fewer females, less right-sided tumors) and outcomes. Despite discordant reports, *BRAF* non-V600E mutated mCRC patients seem to show better survivals compared to those with *BRAF* V600E-mutated tumors, but also to those with wild-type *BRAF* tumors. In a retrospective cohort reported by Jones et al. [30], of 9643 mCRC patients including 208 patients with *BRAF* non-V600E mutation (2.2%), the median overall survival was 60.7 months, 11.4 months, and 43.0 months in patients with *BRAF* non-V600E mutated, *BRAF* V600E mutated and *BRAF* wild-type tumors, respectively (*p* < 0.001). Thus, *BRAF* non-V600E mutated patients could become another subset of patients with distinct therapeutic strategies and their identification at diagnosis might be necessary in the future. Clinical trials dedicated to this subgroup of patients and evaluating the MAPK pathway are already ongoing [31].

The comparative analysis of clinical and pathological characteristics showed that *BRAF* mutated or *RAS* mutated patients have a more aggressive presentation with more synchronous metastases and a lower proportion of resectable disease at diagnosis. Patients with *BRAF* mutation had significantly more right-sided and undifferentiated tumors, showed MSI status, had more infra and supradiaphragmatic adenopathy and peritoneal metastasis, and had less pulmonary disease, which is consistent with previous literature data. MSI patients were more frequently females with right-sided tumors.

In the present study, even if our results are limited to the techniques used with their analytic sensitivity and complexity, we reported a high level of concordance between plasma and tumor *BRAF* mutation status in mCRC patients, especially those with liver metastases. These results validate the clinical implementation of plasma *BRAF* analysis in patients with CRC and liver metastases.

## 4. Materials and Methods 

### 4.1. Study Design and Patients

The RASANC study is a prospective translational study. Patients were eligible if they were 18 years or older and had pathologically confirmed untreated mCRC (with the exception of adjuvant chemotherapy completed ≥6 months prior to enrollment). The exclusion criteria were: No available tumor block, another malignancy in the past five years and medical, sociological, psychological or legal conditions that would not permit the patient to provide informed consent.

The Ile-de-France IV ethics committee approved the research protocol, and all the patients gave their written informed consent. The trial conformed to the Declaration of Helsinki, the Good Clinical Practice guidelines of the International Conference on Harmonization, and relevant French and European laws and directives. The protocol was registered with Clinicaltrials.gov (No: NCT02502656).

### 4.2. Plasma Collection and Tumor Tissue Mutations Analysis

Anonymized blood samples were collected before any study treatment in three 10-mL Streck^®^ tubes and further sent to a centralized laboratory for testing. The tubes were centrifuged 10 min at 1600× *g*, the plasma was recovered and a second centrifugation was performed for 10 min at 6000× *g*. The plasma was then transferred in LoBind tubes (Eppendorf^®^, Hamburg, Germany) and stored at −80 °C until ctDNA analysis.

Tumor *RAS* and *BRAF* mutation status were determined by a French National Cancer Institute (INCa)-approved molecular oncology laboratory and was used to guide the patient care according to standard recommendations. Due to the mutually exclusive nature of the *RAS* and *BRAF* mutations, *BRAF* mutation status was not assessed in all patients with *RAS* mutated tumors. Thus, those with *RAS* mutation in the tumor tissue and without available *BRAF* status were considered as *BRAF* non-mutated patients.

Patients were considered *BRAF* or *RAS* mutated if the tumor tissue and/or plasma was found positive for the mutation of interest.

The choice of first-line chemotherapy was left to the investigators, who were blinded to the results of plasma *RAS* and *BRAF* analysis. Anonymized tumor mutation status reports were centralized.

### 4.3. Data Collection

Baseline clinical and biological data and information about the “standard” and *BRAF* assessment were collected in an electronic case report form (eCRF) using the Cleanweb software (^©^2017 Telemedicine Technologies, Boulogne-Billancourt, France).

The following clinical and biological data were collected: Gender, age, primary tumor location (colon/rectum), tumor differentiation grade, date of primary tumor resection (if done), the TNM classification of primary tumor (if available), date of diagnosis of metastasis (metachronous/synchronous), number and location of metastatic sites, resectability of metastatic disease (yes/potential/never), leukocyte and lymphocyte counts and albuminemia, lactate dehydrogenase, alkaline phosphatase, carcinoembryonic antigen and carbohydrate antigen 19.9 levels.

For the “standard” *RAS* and *BRAF* mutations status assessments, the following information was recorded: Type (biopsy/surgical specimen), origin (primary tumor/metastasis), cellularity, method used, request for *RAS* and *BRAF* mutation testing, receipt of the tumor samples by the laboratory, validation of results by the laboratory and reception of results by the clinician.

### 4.4. RAS and BRAF Mutations Assessment in ctDNA

Circulating tumor DNA was extracted from plasma by using the Maxwell^®^ RSC ccfDNA Plasma Kit (Promega, Lyon, France). DNA was quantified with the Qubit dsDNA HS kit (Thermo Fisher Scientific, Saint Aubin, France) and sequenced using the AmpliSeq Colon and Lung Cancer Panel V2 on an Ion Proton following the manufacturer’s recommendations (Life Technology, Villebon-sur-Yvette, France). The multiplex barcoded ctDNA libraries were generated from 6 µL of plasma and were normalized using the Ion Library Equalizer™ kit. The pooled ctDNA libraries (max. 96) were processed by the Ion Chef™ System for template preparation and chip loading, and were sequenced using the Ion Proton™ System.

The FASTQs sequencing data were aligned to the human genome (hg19) and processed using the Ion Torrent Suite software version 5.0.4.0 with built-in “Somatic-low stringency” parameters allowing the detection of variants with allele frequency of >2%. In parallel, samples were analyzed by the BPER method allowing the detection of variants with allele frequency of <2% [32]. It was implemented within an R package “PlasmaMutationDetector” (https://cran.r-project.org/package=PlasmaMutationDetector) allowing the detection of minimal variants with mutated allele frequency of 0.003 for single-nucleotide and of 0.001 for insertions/deletions with a strong agreement with droplet digital PCR (kappa index 0.90) [32].

In the absence of *BRAF* mutation by NGS, two methylated CRC-specific biomarkers (WIF1 and NPY) were used to determine the presence or absence of ctDNA in plasma by digital droplet PCR [33].

Patients without a *BRAF* mutation by NGS and methylated biomarker by droplet digital PCR were considered to have inconclusive ctDNA results. Patients with at least one mutated gene and/or one methylated biomarker were considered to have conclusive ctDNA results.

Results provided by the package were registered blindly with respect to the *RAS* and *BRAF* tumor status.

### 4.5. Centrally Analyzed Tumor Tissue in Patients with RAS or BRAF Mutation in ctDNA, but Not through Primary Tumor Analysis

When *RAS* or *BRAF* mutation was detected in ctDNA, but not in the corresponding tumor tissue, available remaining tumor tissue sample was analyzed centrally after a second DNA extraction, using the same NGS panel as for plasma analysis. Tumor DNAs were analyzed using the Ion Torrent Suite software. Annotation of variant call format files from the Variant caller plugin was done on a galaxy platform that generates an annotate data file using the pipeline developed for the SAFIR trials (Safir2.4 report tool version 2.4, https://safir2.inserm.fr/OA_HTML/AppsLocalLogin.jsp).

### 4.6. Statistical Analyses

As the principal measure of concordance we used the kappa coefficient, a non-parametric measure of inter-technique agreement introduced by Cohen in 1960 [34]. The Kappa statistic reflects the difference between observed accuracy and expected accuracy (random chance). For example, a kappa value of 0.7 means that agreement is 70% better than by chance alone.

The sample size was determined to assess the concordance measured by the kappa coefficient regarding *RAS* mutation results (ctDNA vs. tumor tissue, considering tumor *RAS* status as the reference). We postulated a 55% frequency of mutated *RAS* and a kappa coefficient of 0.7 (considered satisfactory in Cohen’s classification). With these hypotheses and in order to achieve a precision of ± 0.07 (95% CI: 0.63–0.77) for the kappa coefficient, 425 patients were required (including an expected 5% rate of loss to follow-up and/or non-assessable/unavailable patients).

The kappa coefficient takes into account random chance (agreement with a random classifier), which generally means it is less misleading than simple accuracy. The accuracy was also calculated, but only as a secondary measure of the concordance.

For *BRAF* mutation only, the kappa coefficient and percentage agreement were calculated for the whole population and for patients with conclusive ctDNA results.

The median (interquartile range) and frequency (percentage) were used to describe continuous and categorical variables, respectively. Medians and proportions were compared using the Wilcoxon–Mann–Whitney and chi-square test, respectively, or Fisher’s exact test if appropriate. 

All analyses used SAS software version 9.4 (SAS Institute, Cary, NC, USA). *p*-values < 0.05 were considered statistically significant, and all tests were two-sided.

## 5. Conclusions

This prospective study shows an excellent concordance of a *BRAF* mutation between tissue and plasma in patients with untreated mCRC with the accuracy of 97.3% in the whole population and of 98.5% in patients with conclusive ctDNA results. In patients with *BRAF* mutated in plasma, but wild-type in the tissue, the discordance was due to technical issues in half of the cases. The absence of liver metastasis was the main factor associated with inconclusive ctDNA results. In patients with liver metastasis, the accuracy of *BRAF* mutation detection was 98.6%.

Assessment of the *RAS/BRAF* mutation status in plasma is attracting great interest because of its effectiveness for reducing the delay to obtain molecular results and providing better diagnosis of *BRAF* non-V600E mutated patients. These results, which are consistent with our previous results on *RAS* mutation in this cohort, support ctDNA testing for routine clinical *BRAF* and *RAS* mutation analysis in patients with mCRC and liver metastasis.

## Figures and Tables

**Table 1 cancers-11-00998-t001:** Concordance between the mutational status of *BRAF* in the plasma and tumor tissue.

Population of Interest			*BRAF* Mutation in Plasma Sample		
			Absence *n*	Presence *n*	Total *n*
Whole population ^a^	***BRAF* mutation in tumor tissue**	**Absence *n***	371 (91.6%)	4 (1.0%)	375 (92.6%)
		**Presence *n***	7 (1.7%)	23 (5.7%)	30 (7.4%)
		**Total *n***	378 (93.3%)	27 (6.7%)	405 (100%)
Patients with conclusive ctDNA results ^b^		**Absence *n***	295 (91.3%)	4 (1.2%)	299 (92.6%)
		**Presence *n***	1 (0.3%)	23 (7.1%)	24 (7.4%)
		**Total *n***	296 (91.6%)	27 (8.4%)	323 (100%)

^a^ Kappa coefficient: 0.79 (95% CI: 0.67–0.91); Accuracy: 97.3% (95% CI: 95.2–98.6%); Sensitivity: 76.7% (95% CI, 57.7–90.1%); Specificity: 98.9% (95% CI, 97.3–99.7%). ^b^ Kappa coefficient: 0.89 (95% CI: 0.80–0.99); Accuracy: 98.5% (95% CI: 96.4–99.5%); Sensitivity: 95.8% (95% CI: 78.9–100.0%); Specificity: 98.7% (95% CI: 96.6–99.6%).

**Table 2 cancers-11-00998-t002:** Baseline clinical and pathological characteristics of *BRAF-*mutated patients with conclusive and inconclusive ctDNA results.

Characteristics at Baseline	Conclusive ctDNA Results * N* = 28	Inconclusive ctDNA Results * N* = 6	*p* Value
Sex–*n* (%)			
Missing	0	0	
Male	15 (53.6%)	3 (50.0%)	
Female	13 (46.4%)	3 (50.0%)	1
Age at diagnosis of metastases (years)			
Missing	0	0	
Mean (std)	62.6 (15.3)	66.8 (21.3)	
Median (min–max)	60.5 (28–93)	75 (29–88)	0.3754
Q1–Q3	56–73	56–78	
ECOG PS at diagnosis of metastases–*n* (%)			
Missing	1	0	
Unknown	2	0	
0	6 (25.0%)	1 (16.7%)	
1	13 (52.0%)	3 (50.0%)	
2	6 (24.0%)	2 (33.3%)	1
Primary tumor location–*n* (%)			
Missing	0	0	
Right and transverse colon	19 (67.9%)	4 (66.6%)	
Left and sigmoid colon	6 (21.4%)	1 (16.7%)	
Rectum	3 (10.7%)	1 (16.7%)	1
Primary tumor resection–*n* (%)			
Missing	0	0	
No	18 (64.3%)	1 (16.7%)	
Yes	10 (35.7%)	5 (83.3%)	0.0663
Stage T–*n* (%)			
Missing	0	0	
T0	0	0	
T1	0	0	
T2	0	0	
T3	3 (10.7%)	4 (66.6%)	
T4	8 (28.6%)	1 (16.7%)	
T4	4	1	
T4a	3	0	
T4b	1	0	
Tx	17 (60.7%)	1 (16.7%)	0.0161
Stage N–*n* (%)			
Missing	0	0	
N0	2 (7.1%)	2 (33.3%)	
N1	2 (7.1%)	2 (33.3%)	
N1	1	0	
N1a	0	1	
N1b	1	1	
N1c	0	0	
N2	7 (25.0%)	1 (16.7%)	
N2	4	0	
N2a	1	0	
N2b	2	1	
Nx	17 (60.7%)	1 (16.7%)	0.0414
Grade–*n* (%)			
Missing	0	0	
Not known	6	1	
Well differentiated	7 (31.8%)	2 (40.0%)	
Moderately differentiated	10 (45.5%)	3 (60.0%)	
Undifferentiated	5 (22.7%)	0	0.6667
Metastases–*n* (%)			
Missing	0	0	
Synchronous	26 (92.9%)	4 (66.7%)	
Metachronous	2 (7.1%)	2 (33.3%)	0.1347
Resectability of metastases–*n* (%)			
Missing	1	0	
Resectable	2 (7.4%)	1 (16.7%)	
Potentially resectable	8 (29.6%)	4 (66.6%)	
Non resectable	17 (63.0%)	1 (16.7%)	0.1040
Number of metastatic sites			
Missing	1	0	
Mean (std)	1.96 (1.13)	1.17 (0.41)	
Median (min–max)	2 (1–5)	1 (1–2)	0.0593
Q1–Q3	1–2	1–1	
Localization of metastases–*n* (%)			
Missing	0	0	
Liver	25 (89.3%)	0	<0.0001
Lung	8 (28.6%)	0	0.2975
Peritoneum	7 (25.0%)	5 (71.4%)	0.0703
Adenopathy subdiaphragmatic	9 (32.1%)	1 (16.7%)	0.6445
Adenopathy supradiaphragmatic	5 (17.9%)	0	0.5585
Bone	1 (3.6%)	1 (16.7%)	0.3262
Brain	0	0	-
Other metastatic site	0	1 (16.7%)	0.1765
Leukocytes (/mm^3^)			
Missing	2	0	
Mean (std)	9318.2 (3412.6)	7503.3 (2362.6)	
Median (min–max)	9200 (2142–15,700)	7060 (4640–11,820)	0.0746
Q1–Q3	6500–11,500	6780–7660	
Lymphocytes (/mm^3^)			
Missing	2	0	
Mean (std)	1355.3 (423.7)	1532.8 (418.6)	
Median (min–max)	1340.0 (734–2640)	1575.0 (880–2080)	0.3572
Q1–Q3	1040–1580	1290–1797	
Albumin (g/L)			
Missing	8	1	
Mean (std)	33.9 (7.3)	39.6 (7.1)	
Median (min–max)	34.0 (20.3–50)	41.0 (28.5–46.0)	0.1167
Q1–Q3	28.8–37.3	37.6–45	
CEA (ng/mL)			
Missing	5	0	
Mean (std)	1474.0 (4929.2)	16.6 (20.3)	
Median (min–max)	95.2 (1.3–23,755.1)	4.3 (2.0–47.0)	0.0090
Q1–Q3	25.7–818.2	3.8–38.0	
CA19.9 (U/mL)			
Missing	8	0	
Mean (std)	9110.3 (21,102.4)	4949.3 (10,370.6)	
Median (min–max)	2287.3 (2.6–90,216.0)	225.3 (10.0–25,960.0)	0.3613
Q1–Q3	110.9–5737.9	16.2–3259.0	
LDH (x ULN)			
Missing	11	3	
Mean (std)	3.45 (7.33)	0.99 (0.72)	
Median (min–max)	1.57 (0.80–32.66)	0.67 (0.48–1.80)	0.6015
Q1–Q3	1.07–2.87	0.48–1.80	
ALP (x ULN)			
Missing	6	3	
Mean (std)	2.97 (4.56)	0.59 (0.17)	
Median (min–max)	1.48 (0.46–22.03)	0.64 (0.34–0.72)	0.0405
Q1–Q3	0.73–2.98	0.46–0.71	

Abbreviations: ECOG PS, Eastern Cooperative Oncology Group performance status; ULN, upper limit of normal; CEA, carcinoembryonic antigen; CA 19.9, carbohydrate antigen 19.9; LDH, lactate dehydrogenase; ALP, alkaline phosphatase.

**Table 3 cancers-11-00998-t003:** Kappa coefficients and accuracies for *BRAF* mutation between patients with and without liver metastases and primary tumor resection.

**Overall Population (*n* = 405) ***				
		**Primary Tumor Resection**		
		Yes *n* = 211 (95% CI)	No *n* = 194 (95% CI)	All *n* = 405 (95% CI)
**Liver metastases**	Yes *n* = 289	*n* = 126 K = 1 (1–1) Accuracy = 100.0% (93.3–100.0%)	*n* = 163 K= 0.862 (0.729–0.994) Accuracy = 97.5% (93.8–99.3%)	*n* = 289 K = 0.906 (0.814–0.997) Accuracy = 98.6% (96.5–99.6%)
	No *n* = 116	*n* = 85 K = 0.376 (0.0005–0.753) Accuracy = 92.9% (85.3–97.4%)	*n* = 31 K = NA Accuracy = 96.8% (83.3–99.9%)	*n* = 116 K = 0.345 (0.010–0.701) Accuracy = 94.0% (88.0–97.5%)
	All *n* = 405	*n* = 211 K = 0.736 (0.535–0.937) Accuracy = 97.2% (93.9–99.0%)	*n* = 194 K= 0.835 (0.693–0.976) Accuracy = 97.4% (94.1–99.2%)	*n* = 405 Κ = 0.793 (0.674–0.911) Accuracy = 97.3% (95.2–98.6%)
**Patients with Conclusive ctDNA Results (*n* = 323) ****				
		**Primary Tumor Resection**		
		Yes *n* = 149 (95% CI)	No *n* = 174 (95% CI)	All *n* = 323 (95% CI)
**Liver metastases**	Yes *n* = 265	*n* = 111 K = 1 (1–1) Accuracy = 100.0% (96.7–100.0%)	*n* = 154 K = 0.861 (0.727–0.994) Accuracy = 97.4% (93.5–99.3%)	*n* = 265 K = 0.905 (0.813–0.997) Accuracy = 98.5% (96.2–99.6%)
	No *n* = 58	*n* = 38 K = 0.786 (0.383–1) Accuracy = 97.4% (86.2–99.9%)	*n* = 20 K = NA Accuracy = 100.0% (83.2–100.0%)	*n* = 58 K = 0.791 (0.395–1) Accuracy = 98.3% (90.8–100%)
	All *n* = 323	*n* = 149 K = 0.944 (0.834–1) Accuracy = 99.3% (96.3–100.0%)	*n* = 174 K = 0.863 (0.731–0.994) Accuracy = 97.7% (94.2–99.4%)	*n* = 323 Κ = 0.894 (0.802–0.986) Accuracy = 98.5% (96.4–99.5%)

* patients with inconclusive ctDNA results were considered as non-mutated. ** patients with inconclusive ctDNA results were excluded.

**Table 4 cancers-11-00998-t004:** Baseline clinical and pathological data according to the presence of *RAS* and *BRAF* mutations.

Characteristics at Baseline	Overall Population *n* = 407	*BRAF* Mutated Patients *n* = 34	*RAS* Mutated Patients *n* = 244	Wild Type Patients *n* = 129	*p* Value ^1^
Sex—*n* (%)					
Missing	0	0	0	0	
Male	244 (60.0%)	18 (52.9%)	138 (56.6%)	88 (68.2%)	
Female	163 (40.0%)	16 (47.1%)	106 (43.4%)	42 (31.8%)	0.0627
Age at diagnosis of metastases–years					
Missing	0	0	0	0	
Mean (std)	65.7 (12.7)	63.4 (16.2)	67.6 (11.0)	62.7 (14.0)	
Median (min–max)	67 (20–100)	62 (28–93)	68 (30–96)	64 (20–100)	0.0538
Q1–Q3	59–75	56–75	61–76	55–72	
ECOG PS at diagnosis of metastases—*n* (%)					
Missing	8	1	5	2	
Unknown	9	2	2	5	
0	123 (31.5%)	7 (22.6%)	70 (29.5%)	46 (37.7%)	
1	178 (45.6%)	16 (51.6%)	103 (43.5%)	59 (48.4%)	
2	64 (16.4%)	8 (25.8%)	45 (19.0%)	11 (9.0%)	
3	23 (5.9%)	0	17 (7.2%)	6 (4.9%)	
4	2 (0.5%)	0	2 (0.8%)	0	0.0692
Primary tumor location—*n* (%)					
Missing	0	0	0	0	
Right and transverse colon	125 (30.7%)	23 (67.6%)	75 (30.7%)	27 (20.9%)	
Left and sigmoid colon	168 (41.3%)	7 (20.6%)	102 (41.8%)	59 (45.7%)	
Rectum	114 (28.0%)	4 (11.8%)	67 (27.5%)	43 (33.4%)	<0.0001
Primary tumor resection—*n* (%)					
Missing	0	0	0	0	
No	196 (48.2%)	19 (55.9%)	126 (51.6%)	51 (39.5%)	
Yes	211 (51.8%)	15 (44.1%)	118 (48.4%)	78 (60.5%)	0.0540
Stage T—*n* (%)					
Missing	2	0	2	0	
T0	0	0	0	0	
T1	4 (1.0%)	0	3 (1.2%)	1 (0.8%)	
T2	15 (3.7%)	0	11 (4.6%)	4 (3.1%)	
T3	114 (28.1%)	7 (20.6%)	62 (25.6%)	45 (34.9%)	
T4	87 (21.5%)	9 (26.5%)	47 (19.4%)	31 (24.0%)	
T4	49	5	28	16	
T4a	20	3	10	7	
T4b	18	1	9	8	
Tx	185 (45.7%)	18 (52.9%)	119 (49.2%)	48 (37.2%)	0.2849
Stage N—*n* (%)					
Missing	2	0	2	0	
	73 (18.0%)	4 (11.8%)	41 (16.9%)	28 (21.7%)	
N1	69 (17.0%)	4 (11.8%)	38 (15.7%)	27 (20.9%)	
N1	27	1	15	11	
N1a	16	1	10	5	
N1b	20	2	10	8	
N1c	6	0	3	3	
N2	72 (17.8%)	8 (23.5%)	40 (16.5%)	24 (18.6%)	
N2	23	4	13	6	
N2a	22	1	13	8	
N2b	27	3	14	10	
Nx	191 (47.2%)	18 (52.9%)	123 (50.8%)	50 (38.8%)	0.2708
Grade—*n* (%)					
Missing	12	0	9	3	
Not known	60	7	42	14	
Well differentiated	120 (36.2%)	9 (33.3%)	73 (37.8%)	38 (33.9%)	
Moderately differentiated	194 (58.4%)	13 (48.2%)	112 (58.0%)	69 (61.6%)	
Undifferentiated	18 (5.4%)	5 (18.5%)	4 (4.2%)	5 (4.5%)	0.0348
Metastases—*n* (%)					
Missing	0	0	0	0	
Synchronous	307 (75.4%)	30 (88.2%)	191 (78.3%)	86 (66.7%)	
Metachronous	100 (24.6%)	4 (11.8%)	53 (21.7%)	43 (33.3%)	0.0090
Resectability of metastases—*n* (%)					
Missing	9	1	4	4	
Resectable	50 (12.6%)	3 (9.1%)	222 (9.2%)	25 (20.0%)	
Potentially resectable	120 (30.1%)	12 (36.4%)	59 (24.6%)	49 (39.2%)	
Not resectable	22 (57.3%)	18 (54.5%)	159 (66.2%)	51 (40.8%)	<0.0001
Number of metastatic sites					
Missing	9	1	5	3	
Mean (std)	1.57 (0.81)	1.82 (1.07)	1.55 (0.77)	1.54 (0.80)	
Median (min–max)	1 (1–7)	2 (1–5)	1 (1–7)	1 (1–4)	0.3492
Q1–Q3	1-2	1-2	1-2	1-2	
Localization of metastases—*n* (%)					
Missing	0	0	0	0	
Liver	291 (71.5%)	25 (73.5%)	177 (72.5%)	89 (69.0%)	0.7421
Lung	129 (31.7%)	8 (23.5%)	80 (32.8%)	41 (31.8%)	0.5538
Peritoneum	103 (25.3%)	11 (32.4%)	58 (23.8%)	34 (26.4%)	0.5292
Adenopathy subdiaphragmatic	38 (9.3%)	10 (29.4%)	15 (6.2%)	13 (10.1%)	<0.0001
Adenopathy supradiaphragmatic	21 (5.2%)	5 (14.7%)	9 (3.7%)	7 (5.4%)	0.0244
Bone	18 (4.4%)	2 (5.9%)	15 (6.2%)	1 (0.8%)	0.0511
Brain	5 (1.2%)	0	4 (1.6%)	1 (0.8%)	0.6124
Other metastatic site	34 (8.4%)	1 (2.9%)	21 (8.6%)	12 (9.3%)	0.4788
Leukocytes (/mm^3^)					
Missing	27	2	12	9	
Mean (std)	8555.7 (3508.6)	8977.9 (3287.8)	8638.0 (3630.9)	8284.1 (3328.2)	
Median (min–max)	7995 (771–22,910)	8460 (2142–15,700)	8115 (771–22,910)	7470 (2448–22,600)	0.1974
Q1–Q3	6120–10,210	6610–11,480	6175–10,410	5995–9890	
Lymphocytes (/mm^3^)					
Missing	35	2	18	10	
Mean (std)	1852.1 (2583.2)	1388.6 (421.9)	1934.2 (3271.4)	1820.7 (859.2)	
Median (min–max)	1580.0 (115–48,100)	1350.0 (734–2640)	1596.0 (115–48,100)	1665.0 (510–6000)	0.0537
Q1–Q3	1190–2080	1040–1626	1180–2060	1220–2244	
Albumin (g/L)					
Missing	102	9	48	40	
Mean (std)	36.2 (7.0)	35.0 (7.5)	35.7 (6.9)	37.7 (6.9)	
Median (min–max)	37.1 (16–52)	35 (20.3–50)	37.0 (16–49)	39.0 (17–52)	0.1319
Q1–Q3	32–41.9	29.6–41	32–41	34–42.3	
CEA (ng/mL)					
Missing	52	5	28	15	
Mean (std)	566.5 (2146.5)	1172.4 (4410.4)	671.8 (2192.6)	212.7 (623.8)	
Median (min–max)	36.0 (0.7–23,980)	57.0 (1.3–23,755.1)	57.7 (0.7–23,980)	9.5 (0.8–4146)	0.0006
Q1–Q3	7–199	19–196	11.1–274.0	3.9–99.2	
CA19.9 (U/mL)					
Missing	100	8	65	23	
Mean (std)	4593.8 (37,259.1)	8150.1 (19,056.3)	6483.1 (48,462.6)	531.1 (2190.2)	
Median (min–max)	42.0 (0.6–637,000)	940.6 (2.6–90,216)	87.0 (0.6–637,000)	25.0 (0.8–18,935)	<0.0001
Q1–Q3	12.2–534.1	34–5565.7	18–1184	9.1–79	
LDH (x ULN)					
Missing	200	13	112	66	
Mean (std)	1.83 (2.97)	3.1 (6.82)	1.65 (2.06)	1.78 (2.43)	
Median (min–max)	0.98 (0.09–32.66)	1.50 (0.48–32.66)	0.99 (0.09–17.16)	0.94 (0.28–15.09)	0.0147
Q1–Q3	0.78–1.68	1.02–2.13	0.77–1.67	0.76–1.49	
ALP (x ULN)					
Missing	64	7	26	19	
Mean (std)	1.54 (2.00)	2.62 (4.29)	1.64 (1.92)	1.07 (0.88)	
Median (min–max)	0.86 (0.20–22.03)	1.32 (0.34–22.03)	0.93 (0.20–13.68)	0.77 (0.31–5.07)	0.0476
Q1–Q3	0.60–1.72	0.67–2.80	0.60–1.81	0.58–1.11	

^1^*p*-value from a chi² or Fisher’s test. Abbreviations: WT, wild-type; ECOG PS, Eastern Cooperative Oncology Group performance status; ULN, upper limit of normal; CEA, carcinoembryonic antigen; CA 19.9, carbohydrate antigen 19.9; LDH, lactate dehydrogenase; ALP, alkaline phosphatase.

**Table 5 cancers-11-00998-t005:** *BRAF* mutation status according to microsatellite stability/microsatellite instability phenotype.

*BRAF* Status	MSI *n* = 10	MSS *n* = 123	*p* Value
*BRAF* ctDNA—*n* (%)			
Missing	0	3	
No	4 (40.0%)	100 (83.3%)	
Yes	6 (60.0%)	20 (16.7%)	0.0045
*BRAF* tissue—*n* (%)			
Missing	0	17	
No	4 (40.0%)	83 (78.3%)	
Yes	6 (60.0%)	23 (21.7%)	0.0152

Abbreviations: MSS, microsatellite stability; MSI, microsatellite instability.

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
