# Peer review of "BRAF Mutation Status in Circulating Tumor DNA from Patients with Metastatic Colorectal Cancer: Extended Mutation Analysis from the AGEO RASANC Study"

_cancers, 2019, doi:10.3390/cancers11070998_

Round 1

Reviewer 1 Report

This paper demonstrated the utility of ctDNA for BRAF mutational screening in colorectal cancer patients. The results are well-presented and the conclusions of the study are sound. 

Author Response

We thank the reviewer for its comment. 

Following the suggestion, we revised the whole manuscript carefully. 

Finally, to accommodate all potential editing problems and grammar or syntax errors, an editorial check was provided by medical writer.

Reviewer 2 Report

Mas et al reported an AGEO RASANC study on BRAF mutation analysis in ctDNA from metastatic CRC patient. In general, this study provided significant evidence of concordance between tissue and plasma mutation analysis in metastatic colorectal cancer before molecular characterisation for ctDNA mutation being applied in practice. Compared to the traditional tissue biopsy, which represents a static snapshot of a tumor, liquid biopsy has a unique, great advantage to provide real-time information on disease burden, shedding light on tumor evolution over time. The work is publishable with two concerns:

1) This discussion was really not logically flow. It’s easy to get lost for the statistical analysis for a large cohort of patients. To make it clear, table with results is required to present clearly the data and interpret the results.

2) To make the key points clear, the summary for the key findings obtained from the statistical analysis at the end of the discussion section should be included.

Author Response

We thank the reviewer for this comment.

We addressed the concerns highlighted by the reviewer. The abstract, results, and conclusion sections were extensively modified. 

We added some clarity by including more explanation on the definition of conclusive and inconclusive ctDNA results.

We improved the organization of the results. We also specified several points to make reading easier. We removed Table 3 to add some clarity to the results. 

We added the summary for the key findings of our results in conclusion reflecting the reviewer comment. 

In addition, to improve the quality of the English language and to accommodate all potential editing problems and grammar or syntax errors, an editorial check was provided by medical writer.

Reviewer 3 Report

In this paper Mans et al .  compared tissue and plasma BRAF analysis in 425 patients with colorectal cancer. They reported a high concordanc , especially those with liver metastasis. The topic is interesting, the design is well done and the discussion are according to the results observed. However, the paper is difficult to read and I strongly suggest rewriting it, especially the abstract and the results section.

Author Response

We thank the reviewer for this comment and interest in our work.

Accordingly, the abstract and results section were extensively modified to improve the quality of the manuscript.

We improved the organization of the results. We also specified several points to make reading easier. We removed Table 3 to add some clarity to the results. 

To improve the quality of the English language and to accommodate all potential editing problems and grammar or syntax errors, an editorial check was provided by medical writer.

Round 2

Reviewer 2 Report

The authors have addressed the Reviewer's concerns and comments clearly. This work is publishable.